# The Role of DNA Damage and Repair in Idiopathic Pulmonary Fibrosis

**DOI:** 10.3390/antiox11112292

**Published:** 2022-11-19

**Authors:** Jiahui Zhu, Lexin Liu, Xiaodi Ma, Xinyu Cao, Yu Chen, Xiangping Qu, Ming Ji, Huijun Liu, Chi Liu, Xiaoqun Qin, Yang Xiang

**Affiliations:** 1School of Basic Medicine, Central South University, Changsha 410000, China; 2Department of Medical Laboratory, School of Medicine, Hunan Normal University, Changsha 410000, China; 3Department of Physiology, School of Basic Medicine, Central South University, Changsha 410000, China

**Keywords:** DNA damage, DNA repair, idiopathic pulmonary fibrosis, base excision repair, nucleotide excision repair, mismatch repair

## Abstract

The mortality rate of idiopathic pulmonary fibrosis (IPF) increases yearly due to ineffective treatment. Given that the lung is exposed to the external environment, it is likely that oxidative stress, especially the stimulation of DNA, would be of particular importance in pulmonary fibrosis. DNA damage is known to play an important role in idiopathic pulmonary fibrosis initiation, so DNA repair systems targeting damage are also crucial for the survival of lung cells. Although many contemporary reports have summarized the role of individual DNA damage and repair pathways in their hypotheses, they have not focused on idiopathic pulmonary fibrosis. This review, therefore, aims to provide a concise overview for researchers to understand the pathways of DNA damage and repair and their roles in IPF.

## 1. Introduction

Pulmonary fibrosis is the end-stage of interstitial lung diseases (ILDs), which is characterized by the proliferation of fibroblasts and the accumulation of a large amount of extracellular matrix, accompanied by inflammatory damage and tissue structure destruction. ILDs eventually develop into diffuse pulmonary fibrosis and honeycombing, and patients die from respiratory failure. The most common form of pulmonary fibrosis, idiopathic pulmonary fibrosis (IPF), is a progressive disease with a 5-year survival rate of only 20%, reflecting the lack of effective therapies [1]. The etiology of IPF is still unclear; however, several risk factors and predisposing factors have been proposed, including cigarette smoking, viral infections, and aging [2]. In addition to aggression factors, the lung must also have weaknesses that make these risk factors an opportunity.

The respiratory system communicates with the external environment to maintain gas exchange function. The lung’s surface area is about 100 m^2^, which, coupled with its exposure to the external environment, makes it susceptible to exogenous oxidative stress. Meanwhile, the byproduct of cellular metabolism also acts as a source of reactive oxidative species (ROS), affecting the microenvironment of the lung [3]. Although various enzymatic and non-enzymatic antioxidant defense systems present in all lung cell types protect the lung from damage, oxidative stress exerts its negative effects when homeostasis is out of balance, evidenced by altered cellular or extracellular redox status, increased irreversible oxidative modifications in proteins or DNA, mitochondrial dysfunction, and altered expression or activity of NADPH oxidase (NOX) enzymes and antioxidant enzyme systems [4]. During oxidative stress, oxidatively damaged proteins, lipids, and RNA are typically degraded, while DNA base and strand damage should be repaired to restore genomic integrity. If not fixed, they can be mutagenic and oncogenic or impair normal cell physiology and cellular viability.

A previous study revealed that a DNA-damaged burden above a certain threshold might lead to pulmonary fibrosis via acute epithelial injury, indicating that the DNA damage burden was too high and incompatible with life [5]. Furthermore, DNA damage-related events such as macrophages polarization, epithelial cell dysfunction, fibroblast hyperproliferation, epithelial–mesenchymal transition, myofibroblast differentiation, and cellular aging are all factors in the progression of pulmonary fibrosis [6,7]. Under normal physiological conditions, cells initiate a series of DNA repair machinery after DNA damage, and if the repair is insufficient, lesions will have the opportunity to attack the lung tissue.

DNA repair is the process of recognizing DNA damage from an intracellular surveillance system [8], transducing through a series of signals, and then initiating the repair mechanism to restore the integrity of the genome. To maintain genomic stability, there are five major DNA repair pathways in eukaryotes, namely base excision repair (BER), nucleotide excision repair (NER), mismatch repair (MMR), homologous recombination (HR), and non-homologous end-joining (NHEJ) [9]. Different forms of DNA damage are repaired in different pathways. The most common lesions being small base lesions and DNA single-strand breaks (SSBs) and the most toxic lesions being interstrand crosslinks and DNA double-strand breaks (DSBs) [10]. BER mainly repairs base changes and single-strand breaks. Repair of DSBs includes homologous recombination and non-homologous end-joining.

The purpose of this review was to analyze the literature on DNA repair pathways, and to investigate the possible roles of these pathways in pulmonary fibrosis. Moreover, this review aims to discuss the importance of each DNA repair pathway in pulmonary fibrosis and how this information may assist in the development of therapies. Damage is the switch that triggers the repair pathway, so we will not only elaborate on the connection between the repair pathway and pulmonary fibrosis. In addition, we leave some space to describe DNA damage in pulmonary fibrosis to find the cause of pulmonary fibrosis from the perspective of DNA damage and repair.

## 2. DNA Damage in Idiopathic Pulmonary Fibrosis

Although DNA carries genetic information, its chemical stability in physiological conditions is not as stable as we think, and some reactions such as hydrolysis, oxidation, and non-enzymatic methylation also occur, with oxidative DNA damage as the primary type [10]. In addition to its spontaneous disintegration tendency, DNA damage originates from internal and external genotoxic substances or agents. Endogenous toxicants include ROS, NOS, and estrogen metabolites generated by cell metabolism [11]. Exogenous agents are mainly environmental factors, including chemical attacks, ultraviolet rays, and X-rays [12,13] A meta-analysis of observational studies in countries such as the United Kingdom, the United States, and Japan [14] showed that idiopathic pulmonary fibrosis is associated with environmental exposures, including wood, smoking, and metal dust. Other data have also implicated genetic predisposition factors [15], comorbidities [16] (gastroesophageal reflux disease, etc.), and infections [17] (human herpes viruses, hepatitis C virus, etc.) (Figure 1a). They probably cause varying degrees of damage to DNA.

The main types of DNA damage are single-strand (SSBs) or double-strand breaks (DSBs), abasic sites, bulky adducts, modified bases, interstrand/intrastrand crosslinks, or insertion of intercalating agents [14,18,19,20,21,22,23] (Figure 1b). Of the different types of DNA damage, SSBs are the most DNA lesions, which may result from the attack of DNA bases and deoxyribose by ROS or other electrophilic molecules [16]. Damage results in missing or damaged nucleotides on one of the strands and altered 5′ and/or 3′ ends at the lesion site. However, DNA double-strand breaks are less common than SSBs, which seriously threaten DNA integrity as they can lead to chromosome breaks and translocation.

At present, the molecular mechanism of idiopathic pulmonary fibrosis is still immature. From the perspective of DNA damage, we try to link the risk factors of pulmonary fibrosis with the possible results of DNA damage and to find and summarize the possible evidence associated with the development of idiopathic pulmonary fibrosis.

### 2.1. Environmental Exposures

Several environmental exposures that target the pulmonary epithelium increase the risk of IPF. A study has revealed several vulnerabilities associated with increased risk of IPF via a meta-analysis of six case–control studies, including tobacco smoking, agriculture, farming, livestock, wood dust, metal dust, and stone-sand [14]. The most universal of these factors is cigarettes, which we use as an example to illustrate the possible DNA damage mechanism.

Cigarette smoke is a significant environmental pollutant that causes damage to multiple organs of the body. It has apparent carcinogenic, teratogenic, and mutagenic effects, and its damage to the respiratory system is the most serious and widespread. Various toxic substances in cigarette smoke can directly impact cells or indirectly induce their rupture by producing reactive oxygen species or reactive nitrogen species. As early as 1985, Nakayama et al. confirmed that cigarette smoke solution affected DNA strand break by alkali elution method and studied its mechanism. They believed that the DNA damage caused by cigarette smoke might be related to the carcinogens and free radicals contained in the smoke.

On the one hand, the carcinogens contained in cigarette smoke can directly impact DNA, causing fragmentation; on the other hand, there are some substances in the smoke solution that can indirectly impact DNA by producing reactive oxygen species, such as H_2_O_2_, O^−2^, and OH, and reactive nitrogen [18]. They are known to cause DNA damage and apoptosis in lung epithelial cells, which can trigger the release of TGF-β1 [19]. TGF-β1 stimulates ROS production and the epithelial–mesenchymal transition (EMT) [20], a critical step in fibrogenesis. Thus, it can affect the respiratory system and form pulmonary fibrosis.

Furthermore, Weng et al. [21] found that the levels of XPC and OGG1/2, the two key proteins for NER and BER, are significantly lower in lung tissues of mainstream tobacco smoke (MTS)-exposed mice than in filtered air (FA) mice. These factors influence the development of idiopathic pulmonary fibrosis to varying degrees, which we will discuss later.

### 2.2. Genetic Predisposition

Idiopathic pulmonary fibrosis results from multiple heredity and environmental factors, among which researchers have paid increasingly more attention to genetic factors. Genetic variants found in families were also found in some cases of sporadic IPF. According to the current identification results, variants in several genes were identified as being associated with developing pulmonary fibrosis and related to aging, lung surfactant metabolism, immune response, lung mucus structure protection, and others. Among them, the aging-related telomerase gene TERT and the newly identified genes, such as FAM13A, DSP, OBFC1, ATP11A, and DPP9 [22], are related to DNA damage and repair. The mechanistic link of these genetic variants to pulmonary fibrosis is still unclear, but we still use TERT as an example to review essential information and provide some thoughts.

Telomerase prolongs telomere repair so that telomeres will not be lost due to cell division, increasing the number of cell divisions [23]. It has two primary components: a catalytic component, telomerase reverse transcriptase (hTERT), and an RNA component (hTR); the latter provides a template for hTERT to add nucleotides [24]. A study of pulmonary fibrosis focused on mutations in the telomerase gene and followed families with autosomal dominant dyskeratosis congenita, in which pulmonary fibrosis was the predominant symptom. Of the 73 probands screened, 6 (8%) had heterozygous mutations in hTERT or hTR and they were absent in asymptomatic subjects of the same generation. This verified the relationship between the mutant gene and the inheritance of pulmonary fibrosis. Subsequently, to explore the specific process of telomerase gene mutation related to pulmonary fibrosis, the researchers measured telomere length and found that mutant telomerase was associated with short telomeres. To some extent, telomeres shortening to the limit can be regarded as DNA double-strand breaks. Then, short dysfunctional telomeres activate a DNA damage response that leads to cell death or cell-cycle arrest [25]. Therefore, they proposed that the fibrotic lesion in patients with short telomeres is provoked by a loss of alveolar cells, which is followed by a series of interstitial over-repair processes.

### 2.3. Infectious Microbial Agents

Accumulating evidence suggests that employing antimicrobials positively affects the treatment of pulmonary fibrosis and implies a connection between microbial agents and PF [26,27]. Here, we summarize the latest pathogenic mechanisms of microbial etiology and consider their links to DNA damage.

In recent years, especially after the outbreak of SARS-CoV-19 in 2019, the link between viruses and PF has been receiving increasingly more attention. Viruses detected in IPF patients include EBV, CMV, HHV-8, adenovirus, HCV, TTV, HIV, SARS-CoV-2, and MERS-CoV [28,29,30]. The situation of COVID-19 is still tricky, and the symptoms of pulmonary fibrosis after infection also need to be paid attention to. The data suggest that DNA damage could be caused by the pathogenic process. Viruses can selectively harness or abrogate distinct components of the cellular machinery to complete their life cycles [31]; as a result, numerous cellular pathways, including DNA damage response (DDR), are manipulated. Upon infection of the host cells, the viral genome is translated into proteins, some of which will enter the nucleus and interact with DNA damage repair-associated players. Several studies have reported interactions between SARS-CoV-2 proteins and human cellular proteins using affinity-purification mass spectroscopy (AP-MS) [32,33,34] and bioinformatics analysis [35]. Once the viruses take effect, there will be an accumulation of single-stranded (ss) and double-stranded (ds) breaks, followed by cell apoptosis. The specific mechanism leading to pulmonary fibrosis is still unclear, but if SARS-CoV-2 infects alveolar epithelial cells, apoptosis can be a key starting point for the occurrence of pulmonary fibrosis.

### 2.4. Comorbidities

GERD is a condition of clinical gastroesophageal reflux disease and esophageal mucosal damage caused by excessive exposure to gastric juice in the gastroesophageal cavity [36]. The clinical manifestations are throat, stomach, and respiratory tract symptoms, and we focus on the changes in pulmonary inflammation and interstitial fibrosis caused by repeated inhalation of gastric contents under long-term bed rest. GERD is not a direct cause of idiopathic pulmonary fibrosis, but in patients with chronic lung disease, gastroesophageal reflux promotes the pathological phenomenon of pulmonary fibrosis. The incidence of newly diagnosed GERD is continuously increasing in IPF patients; meanwhile, proton pump inhibitors (PPI) have been associated with a protective effect against IPF-related mortality [37].

GERD may cause chronic microaspiration that leads to repeated subclinical lung injury, which leads to pulmonary fibrosis. Refluxate can further reach the upper part of the esophagus and travel to the airways through microaspiration and eventually to the lungs where it promotes injury through ROS formation and inflammation [38]. The biological consequences of DNA oxidative damage are closely related to human health and certain diseases. Most of the literature involves tumors and aging; however, its mechanism in pulmonary fibrosis is unclear. At least some lines of evidence convincingly showed that DNA damage generated internally or induced by oxidative stress plays a vital role in the development of pulmonary fibrotic diseases, as reviewed elsewhere. Currently, the most concentrated mechanism is the regulation of P53 in the whole process, which is the vital regulator in cell apoptosis, aging, differentiation, etc. ROS-induced DNA damage can trigger p53-dependent apoptosis that may be important in driving a fibrogenic response [39]. In more detail, p53 participates in sensing oxidative DNA damage and modulates BER function in response to persistent ROS stress [40]. No matter how DNA damage is sensed and involved in the development of idiopathic pulmonary fibrosis, the repair is the outcome described in detail below.

## 3. DNA Repair and Its Effects on Idiopathic Pulmonary Fibrosis

### 3.1. Base Excision Repair

A common feature of all risk factors for pulmonary fibrosis development is inflammation, which leads to oxidative DNA damage. As mentioned above, acting as a bridge for the exchange of internal metabolic wastes with fresh air outside, the healthy lung is vulnerable to internal and external oxidative stress, causing oxidative DNA damage. Some lines of evidence convincingly showed that ROS generated internally or induced by environmental exposure plays a vital role in the development of pulmonary fibrosis, as reviewed elsewhere. Bleomycin-induced pulmonary fibrosis is associated with significant increases in reactive oxygen species and oxidized DNA levels [41]. Researchers have detected increased oxidative DNA damage in patients with IPF, silicosis, and asbestos [41,42]. Base excision repair (BER) corrects DNA damage from oxidation, alkylation, and deamination. Such base changes distort the DNA helix very little [43]. Additionally, several BER proteins participate in the single-strand break repair (SSBR) pathway [44], which is essential for maintaining genomic stability.

Typically, BER is initiated by the DNA glycosylases recognition and removing of damaged bases, resulting in the formation of apurinic/apyrimidinic (AP) sites. Immediately after the removal of the bases by a DNA glycosylase, the AP endonuclease incises the DNA backbone 5′ to the abasic site, producing a strand break with a priming 3′-OH group and a non-conventional 5′-deoxyribose phosphate (dRP) [45]. The dRP can be incised by a dRP lyase, and the 3′-abasic unsaturated product of AP lyases is hydrolyzed by AP endonucleases. After three steps of cutting the damaged nucleotides, the gaps in the DNA are filled by DNA polymerase. The repair synthesis step then branched into two pathways, the single-nucleotide-patch BER (SP-BER) and the long-patch BER (LP-BER). In the former case, a single dNMP is incorporated into DNA, and the gap in DNA is sealed by a DNA ligase [46]. In the LP-BER pathway, 2–20 dNMPs displace a stretch of old DNA into a flap structure. As a result, a DNA nick is created, followed by ligation (Figure 2).

Most BER enzymes have been characterized, such as DNA glycosylases, AP endonucleases, DNA polymerases, DNA ligases, and other proteins [46]. In this section, major BER enzymes are described in more detail, with an emphasis on the recently available information related to the pathogenesis and therapies of pulmonary fibrosis.

#### 3.1.1. DNA Glycosylases

DNA glycosylases recognize and excise lesions, which initiates the base excision repair pathway. Six oxidized DNA base-specific glycosylases have been characterized in mammalian cells: MYH, NTHL1, OGG1, NEIL1, NEIL2, and NEIL3 [45]. Given the prominence of oxidative damage in the lung, BER may be paramount for pulmonary fibrosis pathogenesis. The experimental results, which focused on DNA glycosylases in BER, are summarized as follows.

##### OGG1

Given that oxidative damage is often evident in lung disease, the base excision repair pathway targeting oxidative DNA damage is particularly essential in the pathogenesis of pulmonary fibrosis. The primary target of ROS in DNA is guanine, which has the lowest oxidation potential among DNA bases. Among the oxidation products of guanine, 8-hydroxyguanine (7,8-dihydro-8-oxoguanine, 8-oxoG) is the most abundant in DNA and RNA, and its accumulation in DNA is considered to be one of the best biomarkers of oxidative stress [47]. 8-oxoG is mutagenic and capable of pairing with adenine in its “syn” conformation [48]. 8-oxoG must therefore be removed to maintain genome integrity, and this occurs primarily through the DNA base excision repair (OGG1-BER) pathway initiated by 8-oxoG DNA glycosylase 1 (OGG1) [49]. It has been shown that, in addition to the base excision repair function, OGG1 binds 8-oxoG with high affinity, which then interacts with GTPases of the Ras family [50] to catalyze the exchange of GDP with GTP, thereby acting as a guanine nuclear exchange factor. OGG1-BER activates the Rac1 GTPase involved in cellular redox balance and induces the polymerization of smooth muscle actin into stress fibers [45]. Thus, the data may provide a clue to understanding a mechanism by which ROS-induced DNA damage and OGG1-BER of oxidative DNA damage play roles in pulmonary fibrosis.

OGG1 modulates TGF-β1-induced cell transformation and aggravates p-Smad2/3, at least partly by interacting with Smad 7 in the process of pulmonary fibrosis [51]. Although OGG1 participates in DNA damage repair and protects against cell injury, it leads to excessive tissue repair and promotes myofibroblast formation under pathological conditions. In Ogg1^−/−^ mice, bleomycin-induced pulmonary fibrosis was relieved [51], and their findings indicated that OGG1 is a potential molecular target for treating pulmonary fibrosis. Thus, the role of OGG1 in DNA repair prompted the development of OGG1 inhibitors, including O8, SU0268, and TH5487. Among them, TH5487 was used to prevent tumor necrosis factor-α-induced inflammatory response in murine lungs by blocking DNA-OGG1 interactions at the guanine-rich promoter regions of inflammatory-related genes. Ling H et al. focused on the role of TH5487 in pulmonary fibrosis and found that TH5487 attenuated bleomycin-induced fibrosis in WT mice through inhibition of OGG1 expression [52]. The present findings collectively support the development of TH5487 as a promising reagent for the treatment of pulmonary fibrosis.

However, substantial evidence also convincingly confirms that OGG1 led to a reduction in pulmonary fibrosis development. OGG1 is essential for the remission of pulmonary fibrosis in mice exposed to PM2.5 [30] or crocidolite [31]. Furthermore, overexpression of SIRT3 restored the levels and activity of OGG1 and prevented mtDNA damage to rescue the transformation of lung fibroblasts to myofibroblasts [32], implicating that OGG1 is beneficial in attenuating the development of pulmonary fibrosis. In PM2.5-related studies, AEC2s lacking OGG1 have lower proliferation and high NF-κB activation, which lead to hyperactivation of its downstream inflammatory gene expression and more PM2.5-induced pulmonary fibrosis in vivo.

Compared to these inconsistent results with sources of oxidative stress, we found that different stressors act at various sites and activate different signal pathways. Therefore, OGG1 selects the corresponding signaling pathway to exert its repair function. In addition, repair under physiological conditions is conducive to maintaining DNA integrity, but it is prone to excessive repair under pathological conditions. The degree of OGG1 restoration varies in different lung injury settings; maybe its tissue repair is excessive in bleomycin-induced pulmonary fibrosis. Therefore, we speculate that OGG1 functions in its role to a different extent in diverse lung injury settings.

There are inconsistent effects in both OGG1 and OGG1 inhibitors in pulmonary fibrosis. They need to be investigated further and may provide insight into the mechanism of OGG1 in different injury environments, with a view to targeted clinical application.

##### MUTYH

As mentioned above, 8-oxoG is a major ROS-induced oxidized base lesion in DNA, which causes G:C to T:A transversions in genomic DNA. Under physiological conditions, the base excision repair (BER) systems correct it, which are initiated by DNA glycosylases. MUTYH is also a DNA glycosylase enzyme of the BER system; Unlike OGG1, it is responsible for removing an adenine mispaired with 8-oxoG and leaving an apurinic/apyrimidinic site (AP-site) [53]. This site produces a single-strand break (SSB), and will be further repaired [54]. Two MUTYH forms (MUTYH 1 (CCDS41320.1, p57) and MUTYH 2 (CCDS41322.1, p60)) exist in human cells and are encoded by different transcripts. MUTYH 1 has a mitochondrial targeting signal (MTS) at its N-terminus, which is mainly localized in the mitochondria, whereas MUTYH 2 lacks the MTS and localizes it to the nucleus [55].

Given that oxidative stress has been demonstrated to be associated with IPF, as a member of DNA repair, MUTYH deficiency could logically be considered to aggravate oxidative damage and increase the risk for IPF. During severe oxidative stress, SSBs caused by MUTYH cleavage in genomic DNA may accumulate and initiate cell death via the BER process. Sun Q et al. demonstrated that MUTYH deficiency was associated with attenuated pulmonary fibrosis in bleomycin-induced mice [56]. Mechanistically, it can be explained as MUTYH deficiency attenuating the buildup of DNA single-strand breaks (SSB) and maintaining mtDNA integrity in pulmonary tissue cells, which contributed to AECII survival. These findings suggest MUTYH inhibition as a new therapeutic approach to protect the lung from fibrosis under severe oxidative stress. Another study turned its attention to genetic polymorphisms; they found that AluYb8MUTYH polymorphism is a common variant in Chinese and Western populations. IPF patients with the homozygous variant (present/present, P/P) show earlier onset and death ages for IPF [55], which provides a genetic basis for disease development in IPF patients.

#### 3.1.2. Apurinic/Apyrimidinic (AP) Endonucleases

The human apurinic/apyrimidinic (AP) endonuclease APE (also known as Hap1, Apex, and Ref-1) is homologous to Escherichia coli exonuclease III [57], which plays a crucial role in the BER pathway. To begin with, APE, with its endonuclease activity, removes an intermediate product of the repair pathway, the AP site. Apart from its AP endonuclease activity, this enzyme also exhibits 3′→5′ exonuclease, phosphodiesterase, 3′-phosphatase, and Rnase H activities [58]. Functionally, APE is not only responsible for the repair of AP sites but also participates in the redox regulation of transcription factor DNA binding activity [59]. Therefore, APE has been identified as a multifunctional protein involved in the progression of many diseases.

Reports on the association of APE and pulmonary fibrosis have not been found, but many researchers focused on APE’s relationship with the epithelial-to-mesenchymal transition (EMT). The DNA base excision repair gene, APE1, is involved in overexpression in a variety of human cancers [60]. It also has been implicated in protection against cell death resulting from oxidative stimuli and is more sensitive to hyperoxia. To some extent, the possible role of APE1 in pulmonary fibrosis can be analyzed. Yang et al. [61] demonstrated that APE1 promotes TGF-β transition, which plays a core regulatory role in the EMT process. However, Sakai Y et al. [62] found that APE1 knockdown upregulated the expression of TGF-β1 and promoted the dissipation of stress fibers in Α549 cells. Despite the controversies, many laboratories are currently working on clinical feasibility with APE1 inhibitors due to the redox regulation of transcription factors of APE. At present, E3330 suppresses EMT and restores the responsiveness to EGFR-TKIs in vitro and in vivo via inhibition of APE1 redox, which blocks the activity of NF-κB and shows anticancer activities. These findings shed light on the future clinical utility of APE1 redox inhibitors to overcome pulmonary fibrosis.

#### 3.1.3. Scaffolding Proteins

In addition to the core protein, some accessory proteins are also involved in the BER. In this part, we summarize the reported scaffolding proteins linked to pulmonary fibrosis, XRCC1 and PARP-1.

##### XRCC1

Human XRCC1 is the first isolated mammalian gene that affects cell sensitivity to ionizing radiation [63]. As a key scaffold protein of the base excision repair pathway and the other single-strand break repair pathway, XRCC1 plays a central role in coordinating the recruitment/dissociation of DNA damage response factors, stabilizing DNA damage and its surrounding environment, and providing binding sites for response factors, facilitating the complete repair process. During the entire DNA repair process, XRCC1 has been shown to interact with three proteins, DNA pol β, PARP, and LIG3 [64]. These interactions may activate or inhibit the partners and then affect the DNA repair pathway. The result of this is the occurrence of some pathological processes.

Currently, researchers are interested in the involvement of XRCC1 in pulmonary fibrosis after the use of chemotherapy drugs. Despite cisplatin’s apparent effects on cancers, its toxicity for developing lung fibrosis is not to be underestimated. Reduced DNA damage was mainly found in IPF patients after cisplatin treatment. Im et al. verified that the mechanism of cisplatin in fibroblasts from IPF patients is to enhance the activity of XRCC1 for repairing damaged DNA and reducing cisplatin genotoxicity, which subsequently protects fibroblasts from cisplatin-induced cell death. When XRCC1 is silenced, it cannot perform the function of DNA repair. As a result, DNA damage promotes and IPF fibroblast viability decreases [65]. Thus, inhibitors targeting XRCC1 in combination with cisplatin appear to be a therapeutic approach to attenuate pulmonary fibrotic changes over the course of treatment in the future. Given that little is known about the mechanism of XRCC1 in pulmonary fibrosis, its clinical therapeutic potential needs to be further verified.

##### Poly (ADP-Ribose) Polymerases

Poly (ADP-ribose) polymerases (PARPs) are enzymes that use NAD+ to modify themselves and other polypeptides with branched polymeric chains consisting of adenosine 5′-(5′-ribose) diphosphate units [66]. PARP1 is the most studied enzyme in the PARP family. It has been reported that PARP-1 is involved in several fibrotic diseases including the heart [67], liver [68], vessels [69], and lungs [70]. In this review, we focus on its mechanism in pulmonary fibrosis. Compared with XRCC1, its findings are abundant. It is involved in DNA repair and apoptosis, and the role it plays depends on the degree of DNA damage. The production of reactive oxygen species contributes to tissue injury during pulmonary fibrosis. It all starts with DNA breaks caused by reactive oxygen species and triggers energy-consuming DNA repair mechanisms to activate PARP1. In general, activation of PARP1 occurs upon the binding of DNA breaks with its N-terminal zinc finger domain, and then necessary DNA repair proteins are gathered. If the damage is more severe, the activation of PARP1 becomes more efficient, as a result, ATP is exhausted, and a situation of cellular dysfunction and death unfolds.

Yang L et al. [71] have reported that Interactomic analysis suggested that PARP1 is a crutial upstream hub protein in the IPF MPC (networkmesenchymal progenitor cell). In conclusion, PARP-1 is involved in pulmonary fibrosis by inducing lung fibroblast activation and increasing the expression of α-SMA [71]. In contrast, PARP-1-deficient mice exhibited reduced pulmonary fibrosis in response to bleomycin-induced lung injury, relative to wild-type controls [72]. These results suggest that PARP1 inhibitors are important for myofibroblast differentiation and the pathogenesis of pulmonary fibrosis. Unlike OGG1, PARP1 exhibits its consistent mechanism in pulmonary fibrosis, that is, PARP-1 inhibition may be significant.

Genovese T et al. [70] demonstrated that treatment with PARP inhibitors reduces inflammation and tissue injury events induced by bleomycin administration in mice. Lucarini L et al. [73] also verified that HYDAMTIQ, a selective PARP-1 inhibitor, could have a therapeutic potential in reducing the progression of signs and symptoms of the disease by decreasing TGF-β expression and the TGF-β/SMAD transduction pathway.

### 3.2. Nucleotide Excision Repair (NER)

Unlike base excision repair, nucleotide excision repair (NER) can eliminate the widest range of structurally unrelated DNA lesions, for example, cyclobutane–pyrimidine dimers (CPDs) and 6–4 pyrimidine–pyrimidone photoproducts (6–4 PPs), which are usually caused by ultraviolet (UV) radiation; intrastrand crosslinks caused by chemotherapy drugs such as cisplatin; and ROS-generated cyclopurines [74]. Considering they are all risk factors for pulmonary fibrosis, in this part, we link this DNA damage with subsequent repair and search for possible mechanisms of pulmonary fibrosis.

The whole nucleotide excision repair is roughly divided into five steps: (1) DNA-damage recognition: In the global genome nucleotide excision repair (GG-NER) pathway, the damage sensor XPC detects the helix-distorting DNA damage that can be recognized by the UV–DDB (ultraviolet (UV) radiation–DNA damage-binding protein) complex. Once the XPC complex binds to the damage, RAD23B separates from the complex. In the transcription-coupled NER (TC-NER) subpathway, RNA pol II indirectly detects the presence of damage with sensing the blocking of transcript elongation. Then, the TC-NER-specific Cockayne syndrome proteins CSA (ERCC8) and Cockayne syndrome protein B (CSB) are recruited by lesion-stalled RNA Pol II, making it possible to repair DNA lesions. (2) DNA-damage verification: From this step, the two subpathways merge: the transcription initiation and repair factor TF verifies the presence of a lesion with its dual DNA helicases, XPB (recruiting TFIIH to DNA damage) and XPD (detecting the damage) [75,76]. (3) Dual incision and gap filling: The structure-specific endonuclease XPF-ERCC1 and XPG incise the damaged strand to form a gap with the precise localization of the complex composed of XPA, XPG, and RPA. Final DNA gap-filling synthesis and ligation are executed by the replication proteins proliferating cell nuclear antigen (PCNA), replication factor C (RFC), DNA Pol δ, DNA Pol ɛ or DNA Pol κ, and DNA ligase 1 or XRCC1–DNA ligase 3 (Figure 2).

Likewise, we explore key proteins in the NER pathway that have been studied concerning the pathogenesis of pulmonary fibrosis.

#### 3.2.1. ERCC1

ERCC1 (excision repair cross complementing group 1) is an essential member of the exonuclease repair family, a vital member of the NER system, and involved in DNA strand cutting and damage recognition. It is expressed in all tumor cells, and the expression levels vary widely. The expression of ERCC1 directly affects the physiological process of DNA repair.

We did not find any articles related to ERCC1 and pulmonary fibrosis, but some typical pathways of pulmonary fibrosis have obvious intersections with ERCC1. Wang J et al. [77] observed the expression levels of EMT phenotypes and ERCC1 (cisplatin-resistant protein) in the neoadjuvant chemotherapy group and simple surgery group. As a result, regardless of the group, ERCC1 expression showed a significant positive correlation with vimentin and a negative correlation with E-cadherin. Although they did not explore it further, based on these fibrosis indicators, it is not difficult to speculate the effect on pulmonary fibrosis. In the future, we need to further study the repair role of ERCC1 and the consequences of repair failure. What is the relationship between these consequences and pulmonary fibrosis is still an open question and if it is solved, there will be a new way to reduce pulmonary fibrosis.

#### 3.2.2. PCNA

In the NER process, PCNA participates in the excision of DNA single strands and the connection of new strands [78]. XPG recruits PCNA to the site of injury, cleaving nucleotides. These proteins constitute a platform for recruiting DNA polδ for subsequent DNA synthesis and ligase substitution. Studies have shown that PCNA also interacts with some BER proteins, such as MYH, XRCC1, and UNG2 [79,80,81,82]. Then, its extensive role is of great significance to the entire process of DNA repair and is naturally the key to some pathological processes. So, we focus on pulmonary fibrosis.

A 2002 study of honeycomb lesions in pulmonary fibrosis showed that the higher PCNA positivity of the hyperplastic epithelium in the honeycomb lesions of IPF indicates that accelerated cell proliferation occurs in these lesions [83]. Tong B et al. showed that pulmonary PCNA protein was upregulated in bleomycin-treated mice [84]. An early study has confirmed that bleomycin promoted the proliferation of fibroblasts [85]. Other data suggested that PCNA is one of the markers of cellular proliferation [86] and excessive proliferation of fibroblasts is involved in the development of IPF [87,88]. Accumulating evidence strongly indicates that DNA repair protein expression is present during bleomycin-induced lung injury. Furthermore, asbestos and ionizing radiation, such as bleomycin, have been shown to injure the lung by inducing overexpression of P53 and PCNA in the lungs [89,90]. Current research has explored some of the mechanisms, that is, bleomycin-induced upregulation of P53 activates P21, which subsequently blocks the ability of PCNA

To activate DNA polymerase delta [91]. The result of this pathway is extensive damage to the lung epithelium. Although the expression of PCNA is finally increased, part of its effect is the abnormal proliferation of fibroblasts, which does not seem to contradict the mechanism of epithelial apoptosis.

### 3.3. Mismatch Repair (MMR)

The mismatch repair (MMR) system maintains genome stability by repairing base mismatches during DNA replication. MMR gene mutations often lead to microsatellite instability (MSI) and eventually lead to the occurrence of various tumors. The MMR of human DNA is mainly carried out by the combination of MutS homodimer and MutL homodimer to form a tetrameric complex (Figure 2) [92]. MSH2, MSH6, MLH1, and PMS2 are the most important genes regulating the human MMR process. Given that there are currently no studies on the relationship between these proteins and the mechanism of pulmonary fibrosis, we summarize the link between defects in overall MMR and pulmonary fibrosis.

When the MMR gene is methylated or germline mutated, the MMR function can be weakened, and the mismatches generated during DNA replication cannot be repaired, resulting in MSI. Microsatellite (MS) DNA alterations indicate an ineffective MMR system. The MS DNA level could be a valuable index to identify potential altered genes that may play a key role in the pathogenesis of pulmonary fibrosis. Ten MS markers located in chromosomes 8p, 9p, 9q, and 17q were used by Vassilakis et al. to investigate genetic alterations in IPF [93]. They reported that 13 out of 26 (50%) IPF patients manifested either MSI or LOH in at least one of the studied markers. This explains the involvement of genes in different cellular pathways [94] (such as cell cycle, apoptosis, or inflammatory response) in the pathogenesis of pulmonary fibrosis from different perspectives at the molecular level. We did not find other concrete proteins involved in this, and we could not explain the changes in MSI in idiopathic pulmonary fibrosis. So, the mechanism of MMR in idiopathic fibrosis remains to be studied further, and there is still a lack of substantial research evidence.

### 3.4. Homologous Recombination (HR)

DNA-double strand breaks (DSBs) are less common than other forms of damage but are far more deleterious. Until now, γ-H2AX is the most commonly used marker to detect DSBs damage, according to its characteristics in parallel with the degree of DNA damage caused by any form [95]. As stated above, bleomycin induces DNA strand breaks in the presence of iron and oxygen, resulting in lung injury and fibrosis. When the DNA-double strand breaks, eukaryotic cells use two main processes to repair: homologous recombination (HR) and end-joining (EJ) [96]. Among them, homologous recombination is a high-fidelity mechanism that requires a homologous sequence as a template to repair [97] and then restores any missing genetic information in the vicinity of the break site. In most cases, this process uses the sister chromatid as the repair template and is therefore restricted to the G2 and S phases [98]. All in all, in contrast to end-joining, HR involves a more significant number of enzymes and is thus comparatively slower but more accurate [97,99].

HR can be divided into three stages: pre-synapsis, synapsis, and post-synapsis. In pre-synapsis, DNA damage is processed to form an extended region of ssDNA to which a binding protein RPA binds. For DNA double-stranded breaks (DSBs), this step involves a complexity of four nucleases (Mre11-Rad50-Xrs2/NBS1 (MRX/N), Exo1, Dna2, Sae2/CtIP) and a helicase activity (Sgs1/bleomycin) [100]. During synapsis, Rad51 and Rad54 generate D-loop and facilitate the transition from DNA strand invasion to synthesis, respectively [101]. In the last step, post-synapsis, the repair pathway is divided into three subpathways and is relatively complex, mainly involving proteins such as bleomycin, BRCA1, BRCA2, Rad51, Rad52, Rad54, Rad55, and PCNA [102,103] (Figure 3).

Recently, most studies indicated that the disruption of this mechanism (homologous recombination deficiency, HRD) plays an important role in the development and progression of cancer [104], since its deficiency will lead to a high degree of genetic instability and the accumulation of genetic mutations. Although there are no related studies directly discussing the development mechanism of HRD in pulmonary fibrosis, we still found some clues, which verified the involved pathways of HR key proteins by studying the resistance of pulmonary fibrosis to radiation. Im J et al. found that IPF fibroblasts sustain less radiation-induced DNA damage than non-IPF lung fibroblasts. Logically, the most lethal event following radiation to lung cells is the induction of DNA double-strand breaks (DSBs), which are subsequently treated by homologous recombination (HR) repair. They also verified that enhanced FoxM1 in IPF fibroblasts leads to increased RAD51 and BRCA2 gene expression, which are essential repair proteins of HR. Therefore, IPF fibroblasts timely correct their DNA faults and remain more resistant to cytotoxic damage [105]. Why cannot lung epithelial cells have such resistance? They discovered that FoxM1, a member of the Forkhead family of transcription factors [106] is activated secondary to FoxO3a suppression in IPF fibroblasts. At the same time, restoration of FoxO3a function sensitizes IPF fibroblasts to radiation-induced cell death and downregulates FoxM1, RAD51, and BRCA2, indicating that selective targets of HRD may contribute to mitigating fibrosis.

### 3.5. Non-Homologous End-Joining (NHEJ)

Non-homologous end-joining is another pathway for DSB repair, which is not as extensive and accurate as homologous recombination. Instead, it does not depend on a template to repair and directly joins the ends. During NHEJ, the Ku70/Ku80 heterodimer recognizes and binds DNA to break ends, and simultaneously recruits proteins (such as DNA-PKcs, Artemis, and Pol µ or λ) to the repair site for processing, and is finally blocked by the Ligase IV–XRCC4–XLF complex [107,108,109,110] (Figure 3). Likewise, changes in the activity of any one of these proteins can affect the entire repair process. As reported by Pucci S et al. [111], different modulations of Ku70/80 DNA-binding activity in human neoplastic tissues were proven to possibly correlate with tumor progression. We summarize some clues about the impact of key protein alterations in this pathway on pulmonary fibrosis.

#### 3.5.1. Ku70/Ku80

Ku70/80 recognizes a large variety of DSBs with different chemistries and structures of the DNA ends. Apart from this, it also participates in telomere maintenance, replication fork arrest/restart, and cytoplasm [112].

In a study of SIRT1 deficiency and regression of pulmonary fibrosis, the authors unexpectedly found that the attenuation of fibroblast accumulation was associated with decreased deacetylation activity on Ku70 and a remarkable decrease in the expression levels of FLIP [113]. It is not difficult to hypothesize that there is a potential link between the changes in the activity of Ku protein and pulmonary fibrosis.

#### 3.5.2. DNA PKcs

DNA-dependent protein kinase catalytic subunit (DNA-PKcs), a member of the PI3K-related family of enzymes, is a master regulator of the non-homologous end joining of DNA repair [114]. It is also integral to anti-viral responses against DNA viruses, NHEJ DNA repair, VD-J recombination, innate DNA sensing, and telomere maintenance [115,116,117]. Considering the relevance of loss of DNA damage repair pathways in disrepair and senescence in fibrotic lungs [118] and the divergent activities of DNA-PKcs in physiological and pathological settings, some studies were to investigate the role of DNA-PKcs in pulmonary fibrosis. Habiel DM et al. demonstrate that loss of DNA-PKcs promotes the senescence of fibroblasts in IPF, but additional studies are warranted to further explore the DNA-PKcs-dependent mechanisms [118]. Mesenchymal progenitor cells and their fibroblast progeny derived from IPF patients showed a loss of transcripts encoding for DNA damage repair components mediated by the DNA protein kinase catalytic subunit (DNA-PKcs) [119].

## 4. Conclusions

In view of all that has been mentioned so far, one may suppose that the pathogenesis of pulmonary fibrosis includes both genetic and environmental factors. Regardless of these kinds of factors, to some extent, most of them lead to DNA damage, such as oxidative DNA damage, DNA fragmentation, and so on. Meanwhile, apoptosis of epitheliums and other cells in pulmonary fibrosis are also the results of DNA damage. As mentioned above, several studies have linked risk factors of pulmonary fibrosis with lesions of DNA. These can be recognized and repaired by DNA damage repair systems during physiological conditions; however, it seems uncertain under pathological conditions. This review aimed to investigate the role of DNA repair pathways in the development of pulmonary fibrosis. As the lung is exposed to unusually high levels of DNA-damaging stimuli, DNA repair is crucial to preventing fibrosis. We summed up the evidence and research gaps in DNA repair and pulmonary fibrosis (Table 1). Moreover, application of inhibitors (TH5487, E3330, and so on) gives us new insights into new clinical drug development. PF is a complex disease, and in this DNA damage repair framework, it is important to obtain a comprehensive view of each variable in the system to make significant progress.

Therefore, given the progression and complexity of pulmonary fibrosis, early prevention of risk factors and early detection are the most essential. As for the diagnosis and treatment of the later stages, we should not only consider some problems caused by DNA repair defects as a whole but also focus on more specific pathways, even on a certain protein with specific environment, to reduce the impact on other repair effects. In summary, given the paucity of current research on the DNA repair aspects of pulmonary fibrosis, the findings we summarize are only a small part of the evidence, but not the whole story. In the future, more exploration is needed; on the one hand, we need to resolve these disputes, and on the other hand, we need to adopt innovative methods to supplement the exploration of the pathological mechanism and develop new clinical drugs.

## Figures and Tables

**Figure 1 antioxidants-11-02292-f001:**
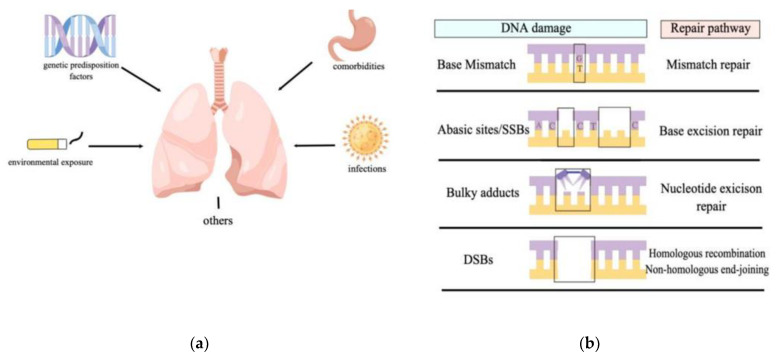
Risk factors of idiopathic pulmonary fibrosis and types of DNA damage and repair. (**a**) Risk factors of IPF: genetic predisposition factors, environmental exposures (wood, smoking, metal dust, etc.), comorbidities (gastroesophageal reflux disease, etc.), and infections (human herpes viruses, hepatitis C virus, etc.) (**b**) Diagram of DNA damage types and corresponding repair pathway: The mismatch repair (MMR) system maintains genome stability by repairing base mismatches during DNA replication. Abasic or AP (apurinic/apyrimidic) sites are cleaved by a DNA glycosylase in the base excision repair (BER) pathway. Nucleotide excision repair (NER) can eliminate the widest range of bulky adducts. When the DNA-double strand breaks, eukaryotic cells use two main processes to repair: homologous recombination (HR) and end-joining (EJ).

**Figure 2 antioxidants-11-02292-f002:**
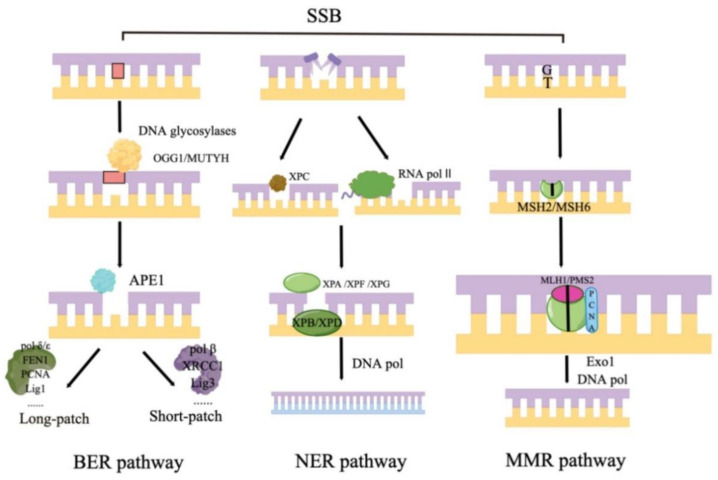
Different pathways of DNA single-strand breaks repair. BER pathway: base damage is recognized by DNA glycosylases that are monofunctional, bifunctional, or Nei-like. APE1 immediately incises the DNA backbone 5′ to the abasic site to generate a gap, which is then filled by a DNA polymerase. The repair synthesis step then branches into two pathways, single-nucleotide-patch BER (SP-BER) and long-patch BER (LP-BER). They all consist of different enzymes. NER pathway: This repair pathway is divided into two branches due to different ways of sensing damage, in the GG-NER pathway, the damage sensor XPC probes the DNA for helix-distorting lesions, and RNA pol II indirectly detects the existence of damage in the other way. Next step, their pathway merges, and XPB and XPD verify the DNA damage. Finally, as with other pathways, a series of proteins perform damage cleavage and nick ligation. MMR pathway: Mispair is recognized by MutS homologs and then recruits MutL homologs and EXO1 exonuclease. Lastly, the daughter strand is excised, and the replicative DNA polymerases fill the gap. (By Figdraw).

**Figure 3 antioxidants-11-02292-f003:**
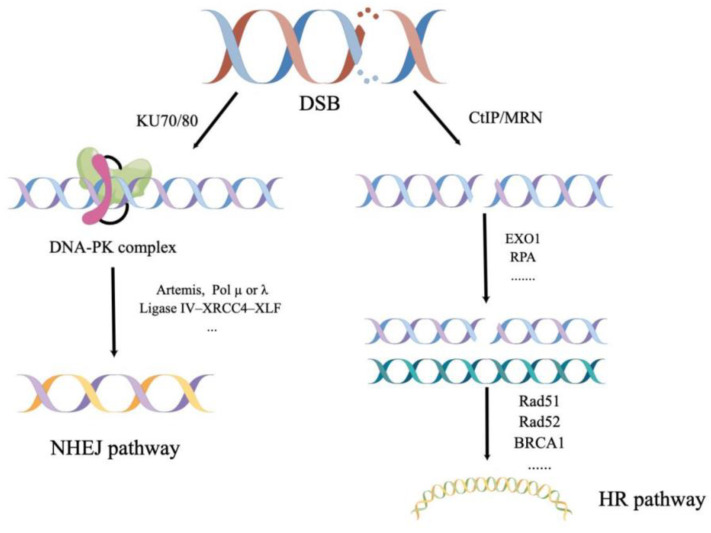
Different pathways of DNA double-strand breaks repair. HR pathway occurs using three steps: extension of ssDNA, a transition from DNA strand invasion to synthesis, and restoration of DNA strand integrity. NHEJ pathway begins with the Ku70/Ku80 heterodimer binding to the ends of the double-strand break (DSB). DNA-PK complex makes this binding even tighter. The ligation step for either strand of the DSB is carried out by the enzymes such as Ligases IV complex shown in the figure. (By Figdraw).

**Table 1 antioxidants-11-02292-t001:** A summary of the evidence and research gaps in DNA repair and pulmonary fibrosis.

DNA Repair Pathways	Research Evidence	Research Gaps
BER *	OGG1 promotes TGF-β1-induced cell transformation/OGG1 led to a reduction in pulmonary fibrosis development	The mechanisms of OGG1 in different injury environments need to be further investigated, with a view to targeted clinical application.
	MUTYH deficiency was associated with attenuated pulmonary fibrosis	
	APE1 promotes TGF-β transition/ APE1 knockdown upregulated the expression of TGF-β1	The clinical feasibility of APE1 in pulmonary fibrosis remains to be investigated.
	XRCC1 deficiency decreased IPF fibroblast viability	Given that little is known about the mechanism of XRCC1 in pulmonary fibrosis, its clinical therapeutic potential needs to be further verified.
	PARP-1 induced lung fibroblasts activation	PARP-1 inhibitor could have a therapeutic potential.
NER	ERCC1 expression showed a significant positive correlation with vimentin	What is the relationship between ERCC1, PCNA and pulmonary fibrosis is still an open question.
MMR	Ineffective MMR markers were used to investigate genetic alterations in IPF	Whether MMR is successful or not may be involved in the pathogenesis of pulmonary fibrosis, but these have not been confirmed.
HR	Selective targets of HR deficiency may contribute to mitigating fibrosis	Selective targets of HR deficiency may contribute to mitigating fibrosis, but specific targets remain blank.
NHEJ	Decreased deacetylation activity on Ku70 was associated with the attenuation of fibroblast accumulation	The evidence of the association between Ku70 and pulmonary fibrosis is limited and needs to be supplemented.
	Loss of DNA-PKcs promotes the senescence of fibroblasts in IPF	Loss of DNA-PKcs promotes the senescence of fibroblasts in IP, additional studies are warranted to further explore the DNA-PKcs-dependent mechanisms

* BER: base excision repair; NER:nucleotide excision repair; MMR: mismatch repair; HR:homologous recombination; NHEJ:non-homologous end-joining.

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
