# Peer review of "The Role of DNA Damage and Repair in Idiopathic Pulmonary Fibrosis"

_antioxidants, 2022, doi:10.3390/antiox11112292_

Round 1
Reviewer 1 Report
The submitted manuscript entitled „ The role of DNA damage and repair in pulmonary fibrosis” by Jiahui Zhu and collogues reviewed to role of DNA damage in fibrosis. The summed up all important mechanisms involved. The manuscript is well written, very clear, logical. However, I do have some general remarks about the manuscript.
Major points:
Figure 1:
The figure does not add additional information to the descriptions in the text. I even do not facilitate the described mechanisms.
Page 3 fourth second paragraph: Environmental exposures
Here fine and ultrafine particles is missing, even it is mentioned later in the manuscript. Please add this to the paragraph.
Page 5 fourth second paragraph: 2.5. Other risk factors
The authors analyzed the effect of bleomycin. Even bleomycin is a drug to treat neoplasms, bleomycin is more an experimental model to induce fibrosis. Bleomycin is mentioned later on in the manuscript, which is sufficient. I would skip this paragraph.
Page 15: Conclusion
This part should be condensed and more generalized.
Minor points:
Page 2 third paragraph last sentence:
“Damage is the switch that triggers repair, and we will also spend some space on the damage of DNA damage to pulmonary fibrosis.” I do not understand what the authors meant with damage of DNA damage.
Page 2 fourth paragraph:
“Endogenous toxicants include ROS, NOS, and estrogen metabolites generated by cell metabolism”. Afterwards the authors than mention radiation as one example. I think radiation is not a poison, even when interfering with DNA. Please select another theme to sum up these interactions.
Author Response
Response to Reviewer 1 Comments
Dear reviewer:
We feel great thanks for your professional work on our article. As you are concerned, there are several problems that need to be addressed. According to your suggestions, we have made extensive corrections to our previous draft, the detailed corrections are listed below.
Point 1: Figure 1:
The figure does not add additional information to the descriptions in the text. I even do not facilitate the described mechanisms.
Response 1: According to your suggestion, we added some notes below the Figure 1.
“Risk factors of idiopathic pulmonary fibrosis and types of DNA damage and repair. (a) Risk factors of IPF: genetic predisposition factors, environmental exposures (wood, smoking, metal dust, etc.), comorbidities (gastroesophageal reflux disease, etc.), and infections (Human herpes viruses, Hepatitis C virus, etc.) (b) Diagram of DNA damage types and corresponding repair pathway: The mismatch repair (MMR) system maintains genome stability by repairing base mismatches during DNA replication. Abasic or AP (apurinic/apyrimidic) sites are cleaved by a DNA glycosylase in the base excision repair (BER) pathway. Nucleotide excision repair (NER) can eliminate the widest range of bulky adducts. When DNA-double strand breaks, eukaryotic cells use two main processes to repair: homologous recombination (HR) and end-joining (EJ).”
Point 2: Page 3 fourth second paragraph: Environmental exposures
Here fine and ultrafine particles is missing, even it is mentioned later in the manuscript. Please add this to the paragraph.
Response 2: Taking your comments and those of other reviewers into consideration, we reviewed relevant literatures and re-summarized the possible environmental exposures. “A meta-analysis of observational studies in countries such as the United Kingdom, the United States, and Japan[14] showed that idiopathic pulmonary fibrosis is associated with environmental exposures, including wood, smoking, metal dust, etc.”
Point 3: Page 5 fourth second paragraph: 2.5. Other risk factors
The authors analyzed the effect of bleomycin. Even bleomycin is a drug to treat neoplasms, bleomycin is more an experimental model to induce fibrosis. Bleomycin is mentioned later on in the manuscript, which is sufficient. I would skip this paragraph.
Response 3:Thanks for your valuable advice. According to your suggestion, we deleted this part.
Point 4: Page 15: Conclusion
This part should be condensed and more generalized.
Response 4: “In view of all of that has been mentioned so far, one may suppose that the pathogenesis of pulmonary fibrosis includes both genetic and environmental factors. Regardless of these kinds of factors, to some extent, most of them lead to DNA damage, such as oxidative DNA damage, DNA fragmentation, and so on. Meanwhile, apoptosis of epitheliums and other cells in pulmonary fibrosis are also the results of DNA damage. As mentioned above, several studies have linked risk factors of pulmonary fibrosis with lesions of DNA. These can be recognized and repaired by DNA damage repair systems during physiological conditions, however, it seems uncertain under pathological conditions.This review aimed to investigate the role of DNA repair pathways in the development of pulmonary fibrosis. As the lung is exposed to unusually high levels of DNA-damaging stimuli, DNA repair is crucial to preventing fibrosis. We summed up the evidence and research gaps in DNA repair and pulmonary fibrosis (table 1). Besides, application of inhibitors (TH5487, E3330, and so on) gives us new insights into new clinical drug development. PF is a complex disease, and in this DNA damage repair framework, it is important to get a comprehensive view of each variable of the system to make significant progress.
Therefore, given the progression and complexity of pulmonary fibrosis, early prevention of risk factors and early detection are the most essential. As for the diagnosis and treatment of the later stages, we should not only consider some problems caused by DNA repair defects as a whole but also focus on more specific pathways, even on a certain protein with specific environment, to reduce the impact on other repair effects. In summary, given the paucity of current research on the DNA repair aspects of pulmonary fibrosis, the findings we summarize are only a small part of the evidence, but not the whole story. In the future, more exploration is needed, on the one hand, we need to resolve these disputes, on the other hand, we need to adopt innovative methods to supplement the exploration of the pathological mechanism and develop new clinical drugs.” We condensed this part according to your suggestion.
Point 5: Page 2 third paragraph last sentence:
“Damage is the switch that triggers repair, and we will also spend some space on the damage of DNA damage to pulmonary fibrosis.” I do not understand what the authors meant with damage of DNA damage.
Response 5: “Damage is the switch that triggers the repair pathway, so we will not only elaborate on the connection between the repair pathway and pulmonary fibrosis. Also, we leave some space to describe DNA damage in pulmonary fibrosis to find the cause of pulmonary fibrosis from the perspective of DNA damage and repair.” We have re-written this sentence according to your suggestion.
Point 6: “Endogenous toxicants include ROS, NOS, and estrogen metabolites generated by cell metabolism”. Afterwards the authors than mention radiation as one example. I think radiation is not a poison, even when interfering with DNA. Please select another theme to sum up these interactions.
Response 6: According to your suggestion, DNA damage originates from internal and external genotoxic substances or agents, and radiation is not a poison, so we substituted “poisons” with “agents”.

Reviewer 2 Report
[Antioxidants] Manuscript ID: antioxidants-1909153
Comments to the authors
The review titled “The role of DNA damage and repair in pulmonary fibrosis” aims to provide a concise overview for researchers to understand the pathways of DNA damage and repair and their roles in pulmonary fibrosis.
The review summarizes and discuss the state-of-the art up to the latest information available in the current relevant literature. The manuscript is well written and organized. I only have some minor comments that should be addressed in particular to facilitate the “review take of message”.
Minor comment
I suggest avoiding the acronym “PV” for pulmonary fibrosis since it is used only 5 time. If the authors like to use it, it must be used consistently through the manuscript.
The reference Kakayama et al does not exist. Did the author refer to Nakayama, T., et al., Cigarette smoke induces DNA single-strand breaks in human cells. Nature, 1985. 314(6010): p. 462-4. Kakayama is reported twice in the manuscript, please correct.
Some full stop and several spaces are missing; for example after the “13]” in the sentence “Exogenous poisons are mainly environmental factors, including chemical attacks, ultraviolet rays, X-rays, etc. [12, 13]An epidemiological study of Asia and Europe showed…”.
Paragraph 2: Please add the reference related to “An epidemiological study of Asia and Europe showed”, is it one manuscript or more than one? One epidemiological study?
Section 2.2: Please substitute “The formation of pulmonary fibrosis” with “Pulmonary fibrosis”
Section 2.1: Please, if any, add also some more recent manuscript(s) that have addressed this issue. The references 15 (1985) is the only one available?
Section 2.3: Please replace “The pathogenic process causes us to think more about DNA damage” with “Data that suggest DNA damage be the pathogenic process cause”.
Section 2.5: Please add at the end of the last sentence “Such dysfunction of the repair system gives a chance to pulmonary fibrosis” the term “development”.
Section 3.1.1.1. OGG1: Page 7, replace “vairious" with “various”.
Section 3,2: Replace “usuallly” with “usually”
Section 3.3: Page 11 “…., but the fact fibroblasts….” What are the fact fibroblasts?
Conclusion section: In the conclusion should be useful underline the data available from human studies and focusing on what is needed to clarify the mechanisms involved in the pulmonary fibrosis initiation and development. Further investigations are needed to fill the gaps (and the contrasting data) present in the literature.
To have a clear state of the art, could be useful to include a table that summarize the available data reporting the references associated with.
Please replace e “epitheliums” with “epithelia” or “epithelial cells”
I have no further comments
Author Response
Response to Reviewer 2 Comments
Dear reviewer:
We feel great thanks for your professional work on our article. As you are concerned, there are several problems that need to be addressed. According to your suggestions, we have made extensive corrections to our previous draft, the detailed corrections are listed below.
Point 1: I suggest avoiding the acronym “PV” for pulmonary fibrosis since it is used only 5 time. If the authors like to use it, it must be used consistently through the manuscript.
Response 1: It may be that the version of the manuscript is different after uploading. We are sorry that we didn’t found any “PV” in our manuscript.
Point 2: The reference Kakayama et al does not exist. Did the author refer to Nakayama, T., et al., Cigarette smoke induces DNA single-strand breaks in human cells. Nature, 1985. 314(6010): p. 462-4. Kakayama is reported twice in the manuscript, please correct.
Response 2: We corrected them according to your suggestion.
Point 3: Some full stop and several spaces are missing; for example after the “13]” in the sentence “Exogenous poisons are mainly environmental factors, including chemical attacks, ultraviolet rays, X-rays, etc. [12, 13]An epidemiological study of Asia and Europe showed…”.
Response 3: We corrected them according to your suggestion.
Point 4: Paragraph 2: Please add the reference related to “An epidemiological study of Asia and Europe showed”, is it one manuscript or more than one? One epidemiological study?
Response 4: We reviewed relevant literature and re-summarized the possible environmental exposures. “A meta-analysis of observational studies in countries such as the United Kingdom, the United States, and Japan[14] showed that idiopathic pulmonary fibrosis is associated with environmental exposures, including wood, smoking, metal dust, etc.”
Point 5: Section 2.2: Please substitute “The formation of pulmonary fibrosis” with “Pulmonary fibrosis”
Response 5: Taking your comments and those of other reviewers into consideration, we substitute “The formation of pulmonary fibrosis” with “Idiopathic pulmonary fibrosis”.
Point 6: Section 2.1: Please, if any, add also some more recent manuscript(s) that have addressed this issue. The references 15 (1985) is the only one available?
Response 6: We added a brief introduction to a recent manuscript that addressed tobacco smoke's effect on DNA damage. That is “Furthermore, Weng, M.W. et al. [21] found that the levels of XPC and OGG1/2, the two key proteins for NER and BER, are significantly lower in lung tissues of mainstream tobacco smoke (MTS)-exposed mice than in filtered air(FA) mice. These factors influence the development of pulmonary fibrosis to varying degrees, which we will discuss later.” Also, we cited M.W. et al.’s original work, referenced as [21].
Point 7:Section 2.3: Please replace “The pathogenic process causes us to think more about DNA damage” with “Data that suggest DNA damage be the pathogenic process cause”.
Response 7: We replaced it according to your suggestion.
Point 8:Section 2.5: Please add at the end of the last sentence “Such dysfunction of the repair system gives a chance to pulmonary fibrosis” the term “development”.
Response 8: Taking your comments and those of other reviewers into consideration, we didn’t add this word, but deleted this paragraph.
Point 9:Section 3.1.1.1. OGG1: Page 7, replace “vairious" with “various”
Response 9: We corrected it.
Point 10:Section 3,2: Replace “usuallly” with “usually”
Response 10: We corrected it.
Point 11:Section 3.3: Page 11 “…., but the fact fibroblasts….” What are the fact fibroblasts?
Response 11: We are sorry for this part, we can not find relevant evidence, so rewrite it as “We did not find other concrete proteins involved in this, and we couldn’t explain the the changes in MSI in idiopathicpulmonary fibrosis. So, the mechanism of MMR in idiopathic fibrosis remains to be studied further, and there is still a lack of substantial research evidence.”
Point 12:Conclusion section: In the conclusion should be useful underline the data available from human studies and focusing on what is needed to clarify the mechanisms involved in the pulmonary fibrosis initiation and development. Further investigations are needed to fill the gaps (and the contrasting data) present in the literature.
To have a clear state of the art, could be useful to include a table that summarize the available data reporting the references associated with.
Response 12: Thanks for pointing out this issue. We revised the conclusion and added table 1 before conclusions.
Point 13:Please replace e “epitheliums” with “epithelia” or “epithelial cells”
Response 13: We replaced it.

Reviewer 3 Report
This review provides clear and up-to-date evidence of DNA damage and repair in IPF.
There is only one query. The title, “The role of DNA damage and repair in pulmonary fibrosis, reminds that this review covers a wide range of Fibrotic ILD, including secondary ILDs, but the review is limited to evidences in studied of IPF. Please change “PF” to “IPF” in the title. Otherwise, please add a trial of secondary ILDs in the text.
Author Response
Response to Reviewer 3 Comments
Dear reviewer:
We feel great thanks for your professional work on our article. As you are concerned, there are several problems that need to be addressed. According to your suggestions, we have made extensive corrections to our previous draft, the detailed corrections are listed below.
Point 1: There is only one query. The title, “The role of DNA damage and repair in pulmonary fibrosis, reminds that this review covers a wide range of Fibrotic ILD, including secondary ILDs, but the review is limited to evidences in studied of IPF. Please change “PF” to “IPF” in the title. Otherwise, please add a trial of secondary ILDs in the text.
Response 1: According to your valuable suggestion, we changed “PF” to “IPF” in the title.
